# The Cost of Diets According to Their Caloric Share of Ultraprocessed and Minimally Processed Foods in Belgium

**DOI:** 10.3390/nu12092787

**Published:** 2020-09-11

**Authors:** Stefanie Vandevijvere, Camille Pedroni, Karin De Ridder, Katia Castetbon

**Affiliations:** 1Sciensano, Service of Lifestyle and Chronic Diseases, 1050 Brussels, Belgium; Karin.deridder@sciensano.be; 2Ecole de Santé Publique, Université Libre de Bruxelles, 1050 Brussels, Belgium; camille.pedroni@ulb.ac.be (C.P.); katia.castetbon@ulb.ac.be (K.C.)

**Keywords:** ultraprocessed foods, minimally processed foods, cost of diets, food consumption surveys, Belgium

## Abstract

Background: This study estimated the monetary cost of diets with higher and lower caloric shares of ultraprocessed food products (UPF) and unprocessed/minimally processed foods (MPF) in Belgium for various sociodemographic groups. Methods: Data from the latest nationally representative Food Consumption Survey (FCS) 2014–2015 (*n* = 3146; 3–64 years) were used. Dietary data were collected through two nonconsecutive 24-hour recalls (food diaries for children). Average prices for >2000 food items (year 2014) were derived from GfK ConsumerScan panel data and linked with foods consumed in the FCS. Foods eaten were categorized by their extent of processing using the NOVA classification. The average caloric share (percentage of daily energy intake) of UPF and MPF were calculated. The mean diet cost was compared across the UPF and MPF contribution tertiles, using linear regression. Results: The average price per 100 kcal for UPF was significantly cheaper (EUR 0.55; 95%CI = 0.45–0.64) than for MPF (EUR 1.29; 95% CI = 1.27–1.31). UPF contributed between 21.9% (female adults) and 29.9% (young boys), while MPF contributed between 29.5% (male adolescents) and 42.3% (female adults) to the daily dietary cost. The contribution of MPF to the daily dietary cost was significantly higher for individuals with a higher household education level compared to those with a lower household education level (*p* < 0.01). Adjusted for covariates, the average dietary cost per 2000 kcal was significantly lower for individuals in the highest compared to the lowest tertile for the proportion of daily energy consumed from UPF (EUR −0.37 ± 0.13; *p* = 0.006), and significantly higher for individuals in the highest compared to the lowest tertile for proportion of daily energy consumed from MPF (EUR 1.18 ± 0.12, *p* < 0.001). Conclusion: Diets with a larger caloric share of UPF were significantly cheaper than those with a lower contribution of these products, while the opposite was found for MPF. Policies that improve relative affordability and accessibility of MPF are recommended.

## 1. Introduction

It has been proposed that food processing, in particular the type, intensity and purpose of food processing, may be linked to human health [1,2]. Consumption of ultraprocessed food products (UPF) has been associated with unhealthy dietary patterns [3,4,5] and with overweight and obesity in studies conducted in the US [6], Canada [7] and Europe [8,9]. For instance, a Spanish cohort study reported that middle-aged adult university graduates in the upper quartile of UPF intake were at a significantly higher risk of overweight or obesity compared to those in the lowest quartile of UPF intake [10]. Other cohort studies from European countries found an association between UPF intake and hypertension [11] and between consumption of UPF and cancer [12], respectively. A Spanish prospective cohort study among university graduates found that a higher consumption of UPF (>4 servings daily) was independently associated with a 62% relatively increased hazard for all-cause mortality [13].

UPF are food or beverage products manufactured from constituents extracted from foods or derived from food ingredients with hardly any intact food, which regularly contain flavors, colors and other additives [14]. Ingredients of UPF include versions of oils and fats, flours and starches, sugar and proteins, including those resulting from further processing, such as hydrogenated oils and fats, modified starches, hydrolyzed proteins and crushed or extruded “mixes” of meat offals or remnants [14].

In some industrialized countries such as the United States and Canada, UPF contribute to more than 50% of total daily energy intake [15,16,17]. In Belgium, a recent study using the data from the nutrition survey 2014/15 found that a higher intake of UPF was significantly associated with a lower dietary quality; while a higher intake of unprocessed/minimally processed foods (MPF) was linked to better dietary quality [18]. Belgians obtain on average 30% of their energy intake from ultraprocessed food products, and this percentage is significantly higher among children than adolescents and adults [18].

When people purchase foods, a large set of determinants may affect their food choices: advertising, packaging and labelling; physical accessibility in stores and other settings; perception of health; individual taste; convenience and cultural norms. Among them, the cost of food is an important determinant, especially for those on the lowest incomes [19]. To date, the monetary cost of UPF has been poorly investigated. A recent study from the United States found that, compared to unprocessed and minimally processed foods, UPF had a lower nutrient density, higher energy density and lower per calorie cost (USD 0.55 vs. 1.45/100 kcal). In addition, UPF did not increase in price as much as unprocessed and minimally processed foods over the last 12 years [20]. In Brazil, in 1995, UPF were the most expensive food group, but the price of UPF underwent successive reductions since the year 2000 and was forecasted to become cheaper than MPF by 2026 [21]. Results from the Brazilian Household Budget Survey 2008–2009, including a random sample of 55,970 households, found that mean prices of foods and beverages purchased at supermarkets were 37% lower in comparison to other food stores, and the share of UPF in purchases made at supermarkets was 25% higher than at other food stores [22]. Another study using the same Household Budget Survey showed that the price of UPF (per kg) was inversely associated with the prevalence of overweight and obesity in Brazil, mainly in the lowest socioeconomic status population [23]. A study comparing the Brazilian Household Budget Survey with a similar survey in the UK found that the caloric share of processed foods and UPF in the UK (63.4%) was well over double that of Brazil (27.7%), whereas their cost relative to the remainder of the diet was 43% lower [24].

The aim of this study was to assess the monetary cost of diets according to their caloric shares of UPF and MPF for different sociodemographic population groups in Belgium. No research has been conducted so far on the cost of diets in Belgium. Moreover, internationally, no studies have been found examining the costs of diets with varying shares of UPF or MPF using representative food consumption survey data.

## 2. Materials and Methods

### 2.1. Food Consumption Survey (FCS)

The Belgian 2014/2015 FCS was organized taking into account the EU Menu guidelines published by The European Food Safety Authority [25]. The survey was accepted by the Human Ethics Committee of the University of Ghent and the Privacy Commission (registration number: B670201319129).

Participants provided written informed consent. The survey design and methodology have been described in detail previously [26,27]. In brief, a representative sample of the Belgian population (*n* = 3146; individuals 3–64 years) was randomly selected from the National Population Register using multistage stratified sampling. The study sample comprised 992 children of 3–9 years, 928 adolescents of 10–17 years and 1226 adults of 18–64 years. Food consumption data in adolescents and adults (10–64 years) were collected using two nonconsecutive 24-hour dietary recalls. GloboDiet^©^ (formerly EPIC-SOFT), a 24-hour recall software program standardized at the European level was used and adapted to the Belgian context [28]. GloboDiet^©^ includes a structured procedure to gather very detailed information and quantities of consumed foods, recipes and dietary supplements. Food portion sizes were estimated using household measures (e.g., glasses, cups, spoons, etc.), manufacturer serving sizes and a photo book containing a selection of country-specific recipes in different serving sizes. Food consumption data in children (3 to 9 years old) were collected using two self-administered nonconsecutive one-day dietary diaries followed by a GloboDiet^©^ completion interview with a proxy respondent (parent or legal guardian). The dietary intake data were linked with the nutritional composition of each food item consumed, through the Belgian Food Composition Database NUBEL (including branded foods) and the Dutch Food Composition database (NEVO).

### 2.2. Covariates

Height was accurately measured to 0.5 cm using a stadiometer (type SECA 213) and weight to 0.1 kg using an electronic scale (type SECA 815 and 804, SECA, Hamburg, Germany) during the second home visit. Information on sex, region and household educational level i.e., education level of the person with the highest education level within the household of the participant (higher education long type (qualified as “high”), higher education short type (“medium”), secondary education or lower (“low”)) was gathered using a computer-assisted personal interview during the first home visit (day of the first 24-hour dietary recall). In children (3–9 years), a parent or legal guardian was used as a proxy respondent.

### 2.3. GfK Consumer Panel

Average data (averaged over the entire year 2014) on food prices for >2000 different food items as per the FCS, including fresh products, were received from Europanel based on the GfK ConsumerScan panel data (Gfk, Brussels, Belgium). The ConsumerScan panel was designed as a stratified sample from the population of private households in Belgium and included about 5000 households. The stratification is based on the household characteristics, age and household size of the reference person (i.e., number of people living in the household). The panel members recorded household purchasing behavior (and related shop-visiting behavior) with respect to a broadly defined group of products regardless of the place of purchase. An electronic measuring instrument was used for purchase registration. This instrument consists of a preprogrammed hand terminal with a 4-line display, a simple keyboard and an integrated scanner for barcode scanning. The purchase and visit data recorded by the respondents were sent to the Research Centre via the GSM network.

A template was prepared based on the GloboDiet^©^ based food classification of the FCS taking into account a range of different product characteristics (i.e., fresh/frozen/canned; full/semi skimmed/skimmed; dried/liquid; full sugar/light/no sugar). Average prices were received from Europanel based on this template and linked to the foods and ingredients consumed in the FCS after taking into account factors for edible parts and yield.

For foods in the FCS with missing prices, the price of the most similar food (in terms of nutritional quality) in the same food group was used. For some food items where prices were given by piece, a conversion was made to obtain the price per kg. For out-of-home meals, the costs of their ingredients were taken into account for the purposes of this study as no prices were available for those meals from GfK.

### 2.4. Food Classification

To classify all foods and ingredients consumed in the FCS according to the type, intensity and purpose of industrial food processing, the NOVA classification [2,29] was used. The NOVA classification divides foods into four categories:(1)Unprocessed or minimally processed foods (MPF) (for example: fresh, chilled, frozen, vacuum-packed vegetables and fruits; unsalted nuts and seeds; fresh, dried, chilled, frozen meats, poultry, fish, seafood).(2)Processed culinary ingredients (for example: plant oils; animal fats; sugar, and salt; starches).(3)Processed foods (for example: canned vegetables and legumes (pulses) preserved in brine; tinned whole or pieces of fish preserved in oil; salted nuts; breads when made from flour, water, ferments and salt).(4)Ultraprocessed foods (UPF) (for example: chips; many types of sweet, fatty or salty snack products; ice-cream; chocolates; candies; margarines; soft, carbonated, cola, energy drinks).

A detailed overview of the application of the NOVA classification to the FCS has previously been published [18]. Some food descriptors used within GloboDiet^©^ were especially useful to apply the NOVA classification, such as type of sweetener (sweetened with sugar, with artificial sweeteners or unsweetened), conservation method (canned, frozen, dried, salted, marinated, candied, fresh etc.), conservation medium (in oil, in water, in own juice, in syrup, etc.), production method (home prepared, industrially prepared, artisanal, catering etc.) and added ingredient or flavor (salted or unsalted). Home-prepared dishes and meals were disaggregated into ingredients, and those ingredients were categorized according to the NOVA classification. For some home prepared composite foods, this disaggregation was not possible (e.g., some milk-based desserts, cakes, pies and pastries, some soups and sauces), but they represented only about 0.9% of all foods consumed. Such foods were classified as processed. Alcohol was not categorized using NOVA and was kept as a separate group.

### 2.5. Data Analysis

Analyses were conducted in SAS 9.3 (SAS Institute Inc., Cary, NC, USA). All analyses took the FCS survey design and sampling weights into account. The proportion of daily energy intake from UPF and MPF was calculated from the average over two interview days (record days for children) using the FCS 2014/15. The average price per 100 kcal for all foods consumed in the FCS2014/15 was compared between UPF and MPF.

The average (over two interview/record days) total daily cost (EUR/d) as well as the total cost per 2000 kcal (EUR/2000 kcal) were calculated for different age, sex and household education level groups. In addition, the average contributions (over two interview/record days) of UPF and MPF to the total daily cost were calculated. Kruskal Wallis tests were conducted by age group to compare the average contributions of UPF and MPF to the total daily cost between individuals with different household education levels. Alcohol was included in the cost and food group contribution estimations.

The cost differential (EUR/2000 kcal) between tertiles of the average proportion of daily energy intake from UPF and MPF was assessed using linear regressions adjusting for age group, sex, household education level and region. A *p*-value of <0.05 was considered statistically significant.

## 3. Results

The average percentage of daily energy intake from UPF and MPF for the total Belgian population was 29.9 and 21.3%, respectively. There were no significant differences between men and women and between different socio-economic population groups in regard to UPF consumption. Intake of UPF among children (3–9 years) was significantly higher (33.3% of daily energy intake on average) compared to adolescents and adults (29.2 and 29.6% of daily energy intake on average, respectively). Individuals with high education level consumed a significantly higher proportion of their daily energy from MPF compared to those with lower education levels (Appendix A
Table A1).

The average price per 100 kcal for UPF consumed in the food consumption survey was significantly cheaper (EUR 0.55; 95%CI = 0.45−0.64) than for MPF (EUR 1.29; 95% CI = 1.27−1.31). The average price per 100 kcal for processed culinary ingredients and processed foods was EUR 0.24 (95%CI = 0.21−0.26) and EUR 0.43 (95%CI = 0.42 − 0.44), respectively. 

The average dietary costs per day and per 2000 kcal are given in Table 1 and Appendix A
Table A2 for different population groups. For all age groups, the average total cost per day was significantly higher for males than females, while the average cost per 2000 kcal was significantly higher for females than males except for children and adolescents (Table 1). The average cost per 2000 kcal significantly increased with the education level of the household in males and females and in all age groups except in young boys (Appendix A
Table A2).

When considering the level of processing of foods and ingredients consumed (Figure 1), MPF contributed the most to the daily dietary cost for all age groups (29.5–42.3%), except for adolescents where processed foods were the highest contributor (34.9 ± 1.0% for males and 31.4 ± 1.0% for females). Contribution of MPF to the daily dietary cost was the highest for female adults (42.3 ± 0.8%) and lowest for male adolescents (29.5 ± 0.8%). UPF contributed between 21.9 and 29.9% to the daily dietary cost, and the contribution was the highest for young children (29.9 ± 1.2%). For all age groups, processed foods contributed more to the daily dietary cost than UPF. Alcohol contributed 6.8 ± 0.5% to the dietary cost for male adults and 3.5 ± 0.4% for female adults (Figure 1).

While there was no significant difference between households of different education levels for the contribution of UPF to the daily dietary cost, the contribution of MPF to the daily dietary cost was significantly higher for age groups with a higher household education level compared to age groups with a lower household education level (*p* < 0.01). The contribution of MPF to the daily dietary cost for households of a low education level varied from 29.5 ± 1.0% for children, 28.3 ± 1.0% for adolescents and 36.0 ± 0.9% for adults. The contribution of MPF to the daily dietary costs for households of high education level varied from 34.3 ± 1.0% for children to 33.4 ± 1.0% for adolescents and 39.3 ± 1.1% for adults (Figure 2).

Adjusted for age group, sex, household education level and region, the average dietary cost per 2000 kcal was significantly lower for individuals in the highest compared to the lowest tertile for the proportion of daily energy consumed from ultraprocessed foods (−0.37 EUR/2000 kcal per day; *p* = 0.006). Similarly, the average dietary cost per 2000 kcal was significantly higher for individuals in the middle (0.61 EUR/2000 kcal per day, *p* < 0.001) and upper (1.18 EUR/2000 kcal per day, *p* < 0.001) tertiles compared to the lowest tertile for the proportion of daily energy consumed from minimally processed foods (Table 2).

Adjusted for the same covariates, with every increase of 1% of daily energy intake from UPF, the average cost per 2000 kcal decreased with EUR0.010 ± 0.002 (*p* < 0.001). Similarly, with every increase of 1% of daily energy intake from MPF, the average cost per 2000 kcal increased with EUR0.048 ± 0.004 (*p* < 0.001).

## 4. Discussion

The cost of healthy diets is widely acknowledged as one of the key factors relating to the healthiness of population diets [30,31], especially for those on lower incomes [19]. This study analyzed for the first time the costs of diets with varying shares of UPF and MPF for different socio-demographic groups in Belgium using representative food consumption survey and food prices data. It has previously been found that higher consumption of ultraprocessed food products was associated with lower diet quality in Belgium [18].

Generally, diets with a higher proportion of energy from UPF are significantly cheaper than diets with a lower proportion of energy from these foods, while the opposite is found for the caloric share of MPF products. Dependent on the age group, between one fifth and a third of the daily monetary cost is spent on UPF products. Households with higher education levels spend significantly more on MPF compared to households with lower education levels. For UPF, no such differences among education level groups were found. This is related to the fact that a previous study did not find significant differences in the proportion of daily energy consumed from UPF between different education levels in Belgium, while significant differences were found for MPF consumption [18]. In other high-income European countries such as Norway [32] or France [33], UPF consumption has been found to be inversely associated with socio-economic position.

A recent study found that increases in the sales of UPF in both high as well as low and middle income countries are closely linked with the industrialization of food systems, technological change and globalization, including growth in the market and political activities of transnational food corporations, as well as inadequate policies to protect population nutrition [34].

Some countries such as Brazil [35,36] and Uruguay [37] officially incorporated UPF in their food-based dietary guidelines. In addition, some countries, such as Chile [38], implemented policies to restrict the marketing of junk food products to children, and other countries, such as Mexico [39,40] and Hungary [41], introduced fiscal measures, such as a tax on junk food. The tax in Mexico has already demonstrated a positive effect with significant declines in the purchases of several categories of UPF [39]. Based on the results from this study, pricing policies would be recommended to reduce the relative affordability of diets with a high caloric share of UPF and improve relative affordability of diets with a high caloric share of MPF in Belgium. Additionally, other structural nonfiscal interventions (i.e., whole food reformulation rather than nutrient-specific reformulation, food procurement policies) will be required to improve accessibility to convenient and affordable minimally processed foods and meals [42].

Strengths of this study include the use of nationally representative nutrition survey data, and the use of a comprehensive set of data on average food prices for Belgium over the same year as the food consumption survey was conducted. The limitations include the absence of data on household income in the FCS and lack of data on prices for out-of-home meals. For these meals, the costs of ingredients were taken into account for the purposes of this study. A consideration of the cost of these meals rather than their ingredients may substantially impact the costs of the diets of the population. We used the average consumption and dietary cost over two interview/record days rather than the usual dietary intakes and dietary costs, as individual usual intakes cannot be calculated using the Statistical Program to Assess Dietary Exposure (SPADE), which is used to calculate the proportion of the population meeting dietary guidelines [43]. In addition, average food prices were used, and so the variation in prices across different supermarkets, regions, lower- and higher socio-economic areas was not taken into account in this study.

Food prices are a complex concept, and there are many factors influencing food prices including political, economic, socio-cultural and environmental factors at a local, national and international level. Price is only one barrier to healthy eating. Other key influences are taste, traditions, convenience, knowledge and cooking skills.

## 5. Conclusion

The average price per 100 kcal for UPF was significantly cheaper than for MPF. The contribution of MPF to the daily dietary cost was significantly higher for individuals with a higher household education level compared to those with a lower household education level. Diets with a larger share of UPF were significantly cheaper than those with a lower share of these products, while the opposite was found for MPF. Policies that improve relative affordability and accessibility of MPF are recommended.

## Figures and Tables

**Figure 1 nutrients-12-02787-f001:**
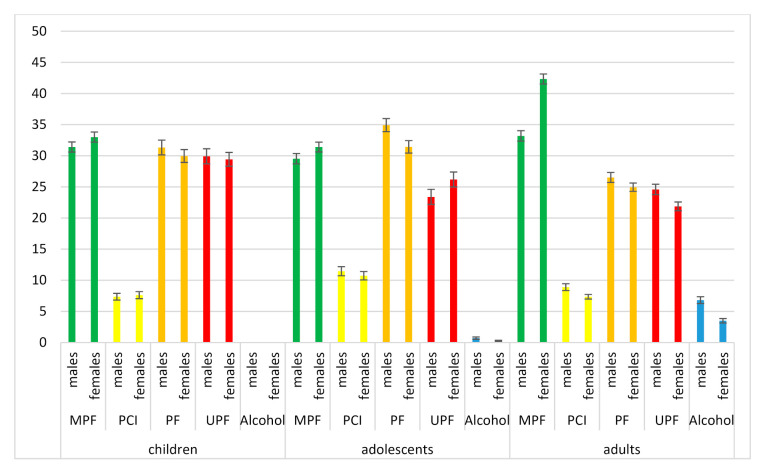
Contribution (average % ± SE) of unprocessed/minimally processed foods (MPF), processed culinary ingredients (PCI), processed foods (PF), ultraprocessed foods (UPF) and alcohol to the cost of diets (total cost/day) by sex and age group.

**Figure 2 nutrients-12-02787-f002:**
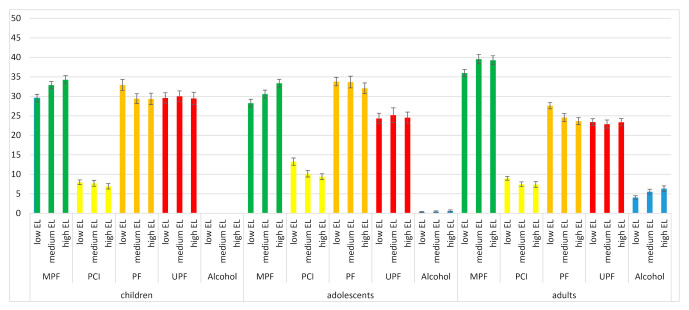
Contribution (average % ± SE) of unprocessed/minimally processed foods (MPF), processed culinary ingredients (PCI), processed foods (PF), ultraprocessed foods (UPF) and alcohol to the cost of diets (total cost/day) by age group and socio-economic status (education level (EL) of the household).

**Table 1 nutrients-12-02787-t001:** Average total cost per day and average cost per 2000 kcal (including alcohol) for different age groups by sex in Belgium (Belgian national nutrition survey 2014/15).

Age Group	Sex		Average Cost	95% CI
**Children** **(3–9 years, *n* = 992)**	Males	EUR/day	4.24 *	4.08–4.41
		EUR/2000 kcal	5.41	5.27–5.54
	Females	EUR/day	3.90	3.77–4.04
		EUR/2000 kcal	5.51	5.35–5.66
**Adolescents** **(10–17 years, *n* = 928)**	Males	EUR/day	5.43 *	5.23–5.64
		EUR/2000 kcal	5.35	5.24–5.45
	Females	EUR/day	4.57	4.42–4.72
		EUR/2000 kcal	5.53	5.40–5.66
**Adults** **(18–64 years, *n* = 1226)**	Males	EUR/day	7.40 *	7.13–7.67
		EUR/2000 kcal	6.58 *	6.42–6.74
	Females	EUR/day	5.67	5.50–5.84
		EUR/2000 kcal	7.15	6.97–7.33

* *p* < 0.05 for comparing males and females. SE: standard error.

**Table 2 nutrients-12-02787-t002:** Cost differential (EUR/2000 kcal) between diets with higher and lower proportions of energy intake (tertiles based on average over two interview/record days) from ultraprocessed foods (UPF) and minimally processed foods (MPF), adjusted for age group, sex, household education level (EL) and region.

	Ultraprocessed Food Products				Unprocessed/Minimally Processed Foods		
Parameter	Estimate	SE	*p*	Parameter	Estimate	SE	*p*
UPF 2 medium %E	0.12	0.13	0.33	MPF 2 medium %E	0.61	0.11	<0.0001
UPF 3 highest %E	−0.37	0.13	0.006	MPF 3 highest %E	1.18	0.12	<0.0001
UPF 1 lowest %E	(ref)			MPF 1 lowest %E	(ref)		
Sex: female	0.43	0.09	<0.0001	Sex: female	0.46	0.09	<0.0001
Sex: male	(ref)			Sex: male	(ref)		
Age group: children	−1.47	0.08	<0.0001	Age group:children	−1.43	0.08	<0.0001
Age group: adolescents	−1.46	0.08	<0.0001	Age group:adolescents	−1.44	0.08	<0.0001
Age group: adults	(ref)			Age group:adults	(ref)		
Household EL: medium	0.30	0.11	0.005	Household EL: medium	0.27	0.10	0.009
Household EL: high	0.34	0.12	0.0006	Household EL: high	0.26	0.12	0.030
Household EL: low	(ref)			Household EL: low	(ref)		
region 2: Brussels	0.18	0.16	0.27	region 2: Brussels	0.09	0.1	0.55
region 3: Wallonia	−0.06	0.09	0.53	region 3: Wallonia	−0.08	0.09	0.39
region 1: Flanders	(ref)			region 1: Flanders	(ref)		

SE: standard error; Ref: reference category.

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
