# Peer review of "The Cost of Diets According to Their Caloric Share of Ultraprocessed and Minimally Processed Foods in Belgium"

_nutrients, 2020, doi:10.3390/nu12092787_

Round 1
Reviewer 1 Report
Review paper:
The cost of diets according to their caloric share of ultra-processed food products among Belgian children, adolescents and adults
This paper adds valuable information on the affordability of diets considering the role of industrial food processing. The study addresses an important gap in the literature and brings useful insights to understand determinants of ultra-processed food consumption, and for informing policy action and other interventions. Nevertheless, the study lacks in demonstrating its quality in the methods section and it reflects in the presentation of the results. The aim is not aligned with what is presented in the results, and the relevance of analyses focused on specific socio-demographics is not stated in the introduction or discussion. The discussion lacks deep and critical reflection of the determinants of UPF prices and the further impact in the food choices. In particular, the role of food corporations’ logistics and manufacturing using low cost ingredients that are decreasing UPFs prices overtime. I missed discussions about the Belgium food system and what is expected in terms of prices of foods in the country. Finally, it is not clear why the comparison is made between MPF and UPF (but in the results other NOVA groups are presented). Considering that MPF are consumed with culinary ingredients in the form of meals, UPF diets should compared with the price per calories of the MPF and PCI accounted together. See below other comments for each section of the paper.
Title: The title is not aligned with the results presented in the paper (focused on other socio-demographics, not only age).
Abstract: Aim: My understanding is that the aim should include information of the other socio-demographics presented in the results. Methods: I suggest to mention that the FCS is a nationally representative survey. It is not clear how the price indicator was used (i.e. per 2000 kcal). Results: It would be interesting to know the mean caloric share, the mean diet cost and price of UPF/2000 kcal, then present the stratified results. It is difficult to follow the results per age and sex, in particular because it is not clear the relevance to stratify those analysis.
Introduction: Missing the rationale to study different population strata. For instance, why would the authors expect different prices by age and sex, and if so, how it can be used for policy purpose and message tailoring? Is there any available data of prices of foods in Belgium that could support the hypothesis? The authors presented data from different household budget surveys, but did not highlighted the relevance of studying the actual food intake, as well as comparing different levels of UPF consumption. The critics about the affordability of UPF should be stated. Other specific comments:
Line 34: I suggest to update references to the NOVA system to include the recent publication by Monteiro et al 2019 “Ultra-processed foods: what they are and how to identify them”.
Line 42-46: Very detailed information for the Spanish context. I suggest to replace with the recent review by Elizabeth et al 2020 “Ultra-Processed Foods and Health Outcomes: A Narrative Review”.
Line 56: I suggest to make clear if the Belgium study used data from the FCS 2014-15.
Line 68: Should it be $/1000 kcal?
Line 70-76: Suggest to consider the following references: Maia et al 2020 “What to expect from the price of healthy and unhealthy foods over time? The case from Brazil” and Jones et al “Meeting UK dietary recommendations is associated with higher estimated consumer food costs: an analysis using the National Diet and Nutrition Survey and consumer expenditure data, 2008–2012”.
Line 82: The impact of fiscal policies it is not aligned with the objective and looks out of place.
Methods: The data analysis section is particularly weak and did not provide enough information of the analysis presented in the results (e.g. stratification). It is not clear when and why one or two 24-h recall (or record) were used. My understanding is that the number of days to estimate UPF intake does not always match with number of days used to calculate the cost, what are the implications for the robustness of the analysis? Could you please provide the rationale for including BMI as a covariate? Were other socio-demographic characteristics collected in the survey? Why income was not considered in the analysis given its strong correlation with UPF consumption and expenses? Detailed information on how out of home meals (e.g. fast foods) were accounted should be included. Finally, I think that presenting relative prices (UPF/MPF) would be better to compare across different groups. Other specific comments:
Line 103: It is not clear how the self-administered data collection was conducted for children (in particular for the pre-school children). Could you provide more information if the children and adults were from the same household? If yes, what would be the implications for the analysis by household education (e.g. are the estimations being repeated?).
Line 114: Could you please include the thresholds and information if obesity was also identified for adults and how?
Line 126: It is not clear what household size means (e.g. number of people living in the household).
Line 134: Could you provide more information on how the average prices were derived? Do they represent the average per year, was the price deflated?
Line 138: Considering the main aim of the paper, why food processing was not applied to input price of the similar food instead of nutritional quality?
Line 141-144: I suggest to make clear that the NOVA system consider the industrial food processing. It is also redundant the final sentence about degree of processing.
Line 152: Although the authors refer to another publication, could you please provide more information where information on recipes came from?
Line 156: Instead of sugar content, I suggest ‘type of sweetener’.
Line 158: It is not clear what medium refers to.
Line 160: It is not clear what ingredient refers to.
Line 165: Home prepared foods should not be classified as processed foods as by definition those are manufactured products – unless the main component of the dish was a processed food. The % of items that fall on this category are very low, so would not be a problem.
Line 173: Why is relevant to present the information of alcohol separately and not as part of the processed or UP food groups?
Line 177: Could you please provide more information on how the tertiles were calculated?
Results: The results are not appealing for the reader and do not highlight key messages such as the average price of diets according to the level of processing for the overall population. The contribution of UPF to the overall cost of the diet is not very informative as it is directly related to the energy contribution. Comparisons per 2000 kcal are much more informative. Also missing information of the energy contribution of the NOVA groups. In the average diet, UPFs are already cheaper or it becomes more pronounced when more of these foods are consumed (higher tertiles of consumption)? If yes, how the authors later explain it and what is the potential relationship with subgroups consumed? Title of tables should be reviewed to remove information that can be included as footnote, e.g. “including alcohol” on Tab 1, adjustment variables on Tab 2, tertiles based on average over two interview/record days on Tab3. Other specific comments:
Table 1: The range is not informed. Why the comparison is made by sex and not age groups if it is the main aim of the study? The comparison should also be made for the cost/2000 kcal.
Line 196: Analysis for the other NOVA groups are not stated in the methods.
Tab 2: Why the parameter UPG %E is not present in ordinal form?
Discussion: Missing critical discussion of the results based on similar studies and changes in the food system that are increasing the affordability of UPF, in particular in the Belgium context and the differences with low and middle income countries. The flow between the first and second paragraph is not very clear, and it was difficult to understand that the second paragraph referred to actual results of study. The strengths of using food consumption and not budget surveys (as other studies) should be highlighted, as well as the relevance to look at tertiles of consumption. If seasonality was not considered in the study, it should be stated as a limitation. Other specific comments:
Line 245-248: references are not adequate, e.g. ref Monteiro et al 2015 is not the Brazilian dietary guidelines, Fiolet is not the policy publication. For marketing regulation in Chile, is there an academic reference for it?
Line 266-271: limitations are mixed with conclusions
Conclusion: Should be improved to highlight main results and respective implications for research and policy.
Author Response
Answer comments reviewers
Reviewer 1
Review paper:
The cost of diets according to their caloric share of ultra-processed food products among Belgian children, adolescents and adults
This paper adds valuable information on the affordability of diets considering the role of industrial food processing. The study addresses an important gap in the literature and brings useful insights to understand determinants of ultra-processed food consumption, and for informing policy action and other interventions. Nevertheless, the study lacks in demonstrating its quality in the methods section and it reflects in the presentation of the results. The aim is not aligned with what is presented in the results, and the relevance of analyses focused on specific socio-demographics is not stated in the introduction or discussion. The discussion lacks deep and critical reflection of the determinants of UPF prices and the further impact in the food choices. In particular, the role of food corporations’ logistics and manufacturing using low cost ingredients that are decreasing UPFs prices overtime. I missed discussions about the Belgium food system and what is expected in terms of prices of foods in the country. Finally, it is not clear why the comparison is made between MPF and UPF (but in the results other NOVA groups are presented). Considering that MPF are consumed with culinary ingredients in the form of meals, UPF diets should compared with the price per calories of the MPF and PCI accounted together. See below other comments for each section of the paper.
Answer from the authors: We have made some changes to address your comments. Please find the replies to your detailed comments below.
Title: The title is not aligned with the results presented in the paper (focused on other socio-demographics, not only age).
Answer from the authors: We have now adapted the title to a more generic title as follows as indeed the study focuses on different socio-demographic population groups: The cost of diets according to their caloric share of ultra-processed and minimally processed foods in Belgium.
Abstract: Aim: My understanding is that the aim should include information of the other socio-demographics presented in the results. Methods: I suggest to mention that the FCS is a nationally representative survey. It is not clear how the price indicator was used (i.e. per 2000 kcal). Results: It would be interesting to know the mean caloric share, the mean diet cost and price of UPF/2000 kcal, then present the stratified results. It is difficult to follow the results per age and sex, in particular because it is not clear the relevance to stratify those analysis.
Answer from the authors: We replaced ‘age groups’ in the aim in the abstract with ‘socio-demographic groups’ instead. We have added in the abstract that the FCS is a nationally representative survey. We have added in the abstract and in the results the mean price (per 100 kcal) for ultra-processed versus minimally processed foods. The key data on the caloric share of UPF and MPF have previously been published for Belgium and so we prefer not to include those in the abstract. We did elaborate a bit more on it in the results section and added a table in appendix with those data.
Introduction: Missing the rationale to study different population strata. For instance, why would the authors expect different prices by age and sex, and if so, how it can be used for policy purpose and message tailoring? Is there any available data of prices of foods in Belgium that could support the hypothesis? The authors presented data from different household budget surveys, but did not highlighted the relevance of studying the actual food intake, as well as comparing different levels of UPF consumption. The critics about the affordability of UPF should be stated. Other specific comments:
Answer from the authors: We have now adapted the aim more generally to studying the cost of diets according to share of UPP and MPF foods for different socio-demographic groups. In regards to studying the relevance of actual food intake as well as comparing levels of UPF consumption, this has already been done for Belgium, as we also explained in the introduction section and we made reference to the paper already published here:
Vandevijvere S, De Ridder K, Fiolet T, Bel S, Tafforeau J. Consumption of ultra-processed food products and diet quality among children, adolescents and adults in Belgium.Eur J Nutr. 2019 Dec;58(8):3267-3278. doi: 10.1007/s00394-018-1870-3.
So the focus of the current paper is really on the cost estimations and contributions.
Line 34: I suggest to update references to the NOVA system to include the recent publication by Monteiro et al 2019 “Ultra-processed foods: what they are and how to identify them”.
Answer from the authors: We have now added this reference where you suggest.
Line 42-46: Very detailed information for the Spanish context. I suggest to replace with the recent review by Elizabeth et al 2020 “Ultra-Processed Foods and Health Outcomes: A Narrative Review”.
Answer from the authors: If possible we prefer to keep those studies as these are data from cohort studies which is why we used those.
Line 56: I suggest to make clear if the Belgium study used data from the FCS 2014-15.
Answer from the authors: We updated the sentence as follows: In Belgium, a recent study using the data from the latest food consumption survey 2014/15 found that higher consumption of UPF was significantly associated with lower diet quality (lower fruit and vegetable consumption for some age groups, higher salt and saturated fat intake); while the opposite was observed for higher consumption of unprocessed/minimally processed foods (MPF) (Vandevijvere et al, 2018).
Line 68: Should it be $/1000 kcal?
Answer from the authors: No it is correct as it is mentioned in the text as $/100 kcal
Line 70-76: Suggest to consider the following references: Maia et al 2020 “What to expect from the price of healthy and unhealthy foods over time? The case from Brazil” and Jones et al “Meeting UK dietary recommendations is associated with higher estimated consumer food costs: an analysis using the National Diet and Nutrition Survey and consumer expenditure data, 2008–2012”.
Answer from the authors: We have added the first reference and also some text in relation to those in the introduction section. We have not added the second reference since it is not specifically reporting on UPF. There are a lot of other studies on the cost of diets in relation to recommendations that could be referenced.
Line 82: The impact of fiscal policies it is not aligned with the objective and looks out of place.
Answer from the authors: Indeed. We have removed this
Methods: The data analysis section is particularly weak and did not provide enough information of the analysis presented in the results (e.g. stratification). It is not clear when and why one or two 24-h recall (or record) were used. My understanding is that the number of days to estimate UPF intake does not always match with number of days used to calculate the cost, what are the implications for the robustness of the analysis? Could you please provide the rationale for including BMI as a covariate? Were other socio-demographic characteristics collected in the survey? Why income was not considered in the analysis given its strong correlation with UPF consumption and expenses? Detailed information on how out of home meals (e.g. fast foods) were accounted should be included. Finally, I think that presenting relative prices (UPF/MPF) would be better to compare across different groups.
Answer from the authors: We have rewritten the data analysis section. To clarify, for all analyses two 24-hour recalls (or two records for children) have been used. We also removed BMI as a covariate in the regression analyses and updated the results of those regressions accordingly in Table 2. We agree that it would have been very important to have data on household or individual income, but unfortunately the Belgian food consumption surveys only collect data on education level and not on income. So we do not have data on income available for this study. We added this as a limitation of the study.
We added under the section on GfK consumer panel data that for out-of-home meals, the costs of their ingredients were taken into account for the purposes of this study as no prices were available for those meals from GfK.
Other specific comments:
Line 103: It is not clear how the self-administered data collection was conducted for children (in particular for the pre-school children). Could you provide more information if the children and adults were from the same household? If yes, what would be the implications for the analysis by household education (e.g. are the estimations being repeated?).
Answer from the authors: No households were selected but individuals, so the children and adults were not from the same household in this survey. In children (aged 3 to 9 years old) a parent or legal guardian was used as a proxy respondent. To collect dietary intake in children, a parent (or legal guardian) was asked to keep a food diary on two non-consecutive days. This was followed by a completion interview with a computerized 24-h recall program. The first completion interview was performed by telephone, or exceptionally by an additional face-to-face interview, before the follow-up home visit. During the follow-up home visit a second 24-h recall was performed in adults and adolescents. In children the second completion interview based on the completed one-day food diary was performed. All food-diaries were open-ended (i.e. no pre-coded food lists) and special pages were available for home-made recipes and dietary supplement intake. In every booklet explanations and examples on how to fill in the diary were provided. We referred to other papers that have reported more in detail on the food consumption survey methods.
Line 114: Could you please include the thresholds and information if obesity was also identified for adults and how?
Answer from the authors: We have actually removed reference to thresholds for both adults and children because we are not using BMI anymore for the purposes of this study as we removed BMI as a covariate from the regression analyses
Line 126: It is not clear what household size means (e.g. number of people living in the household).
Answer from the authors: We have now added that household size indeed means the number of people living in the household.
Line 134: Could you provide more information on how the average prices were derived? Do they represent the average per year, was the price deflated?
Answer from the authors: The household panel of 5000 households register the prices of the food they buy continuously. The average prices are calculated using the average over the year 2014 and weighted for sampling variables. We clarified this in the manuscript.
Line 138: Considering the main aim of the paper, why food processing was not applied to input price of the similar food instead of nutritional quality?
Answer from the authors: This is the most common approach, in reality these foods would also have the same extent of processing. For example if there was no price for white beans, it was assigned the price of red beans. We did not assign prices of very different foods to foods which were lacking the prices.
Line 141-144: I suggest to make clear that the NOVA system consider the industrial food processing. It is also redundant the final sentence about degree of processing.
Answer from the authors: We have clarified that NOVA considers industrial processing and we have deleted the last part of the sentence (i.e. according to their degree of processing)
Line 152: Although the authors refer to another publication, could you please provide more information where information on recipes came from?
Answer from the authors: Description and quantification of both foods and recipes is done by the participants. When participants do not remember or for out-of-home meals, standardized recipes were used. GloboDiet® involves a structured and standardized approach to collect very detailed descriptions and quantities of consumed foods, recipes and dietary supplements. Food portion sizes were quantified using weights, volumes, shapes, thicknesses (e.g. spreads), household measures (e.g. glasses, cups, spoons, etc.), standard food portions (e.g. apples, packages) and a picture book including a selection of country-specific dishes in different portion sizes.
Line 156: Instead of sugar content, I suggest ‘type of sweetener’.
Answer from the authors: We have made this change
Line 158: It is not clear what medium refers to.
Answer from the authors: We have reworded ‘medium’ to ‘conservation medium’. The examples are between brackets.
Line 160: It is not clear what ingredient refers to.
Answer from the authors: We have reworded ‘ingredient’ to ‘added ingredient or flavour’. The examples are between brackets.
Line 165: Home prepared foods should not be classified as processed foods as by definition those are manufactured products – unless the main component of the dish was a processed food. The % of items that fall on this category are very low, so would not be a problem.
Answer from the authors: We prefer to keep this as this was similarly done in a previous paper already published. Indeed the percentage of such items is very low.
Line 173: Why is relevant to present the information of alcohol separately and not as part of the processed or UP food groups?
Answer from the authors: Alcohol is different from non-alcoholic foods and beverages as policy regulations are usually different. Countries usually do not have the same regulations on non-alcoholic beverages and foods and alcohol. Moreover the study is including children and adolescents in addition to adults.
Line 177: Could you please provide more information on how the tertiles were calculated?
Answer from the authors: We feel this is quite obvious and not sure what should be added? The tertiles are calculated based on the average proportion of daily energy consumed from either UPF or MPF.
Results: The results are not appealing for the reader and do not highlight key messages such as the average price of diets according to the level of processing for the overall population. The contribution of UPF to the overall cost of the diet is not very informative as it is directly related to the energy contribution. Comparisons per 2000 kcal are much more informative. Also missing information of the energy contribution of the NOVA groups. In the average diet, UPFs are already cheaper or it becomes more pronounced when more of these foods are consumed (higher tertiles of consumption)? If yes, how the authors later explain it and what is the potential relationship with subgroups consumed? Title of tables should be reviewed to remove information that can be included as footnote, e.g. “including alcohol” on Tab 1, adjustment variables on Tab 2, tertiles based on average over two interview/record days on Tab3.
Answer from the authors: We have now started the results section highlighting the caloric share of UPF and MPF for different population groups and added a table in Appendix. These data have been published previously but are indeed still important to consider when interpreting the key results of this study. So we added them in Appendix.
We have also added a paragraph on the actual average prices of UPF versus MPF consumed in the food consumption survey. We feel it is important to include the contribution of the different NOVA food groups to the total daily cost of the diet. We prefer to keep the full titles of the tables and figures if possible.
Other specific comments:
Table 1: The range is not informed. Why the comparison is made by sex and not age groups if it is the main aim of the study? The comparison should also be made for the cost/2000 kcal.
Answer from the authors: For Table 1 and appendix Table 2 we have actually removed the SE and the min/max as the confidence intervals are more informative. So we have added those instead. The text in the results section does refer to both age and sex for Table 1.
Line 196: Analysis for the other NOVA groups are not stated in the methods.
Answer from the authors: The main groups of interest in this study are MPF and UPF. Some of the figures include all four groups, they have been introduced in the method section under food classification.
Tab 2: Why the parameter UPG %E is not present in ordinal form?
Answer from the authors: We do not understand this comment?
Discussion: Missing critical discussion of the results based on similar studies and changes in the food system that are increasing the affordability of UPF, in particular in the Belgium context and the differences with low and middle income countries. The flow between the first and second paragraph is not very clear, and it was difficult to understand that the second paragraph referred to actual results of study. The strengths of using food consumption and not budget surveys (as other studies) should be highlighted, as well as the relevance to look at tertiles of consumption. If seasonality was not considered in the study, it should be stated as a limitation.
Answer from the authors: There are no similar studies using food consumption survey data to evaluate the cost of diets with higher and lower shares of UPF and MPF. We did cite the relevant literature using household budget surveys or looking at food prices in the introduction. We have added reference to an important paper by Baker et al who recently conducted a comprehensive review on UPF and their drivers. The following was added to the discussion section: A recent study found that increases in the sales of UPF in both high as well as low and middle income countries are closely linked with the industrialization of food systems, technological change and globalization, including growth in the market and political activities of transnational food corporations, as well as inadequate policies to protect population nutrition (Baker et al, 2020). We have added the strength of using food consumption survey data.
Other specific comments:
Line 245-248: references are not adequate, e.g. ref Monteiro et al 2015 is not the Brazilian dietary guidelines, Fiolet is not the policy publication. For marketing regulation in Chile, is there an academic reference for it?
Answer from the authors: We have updated these references. We removed the sentence on France as we do not find the original policy or report.
Line 266-271: limitations are mixed with conclusions
Answer from the authors: We have one paragraph with strengths and limitations before the conclusion so we do not really understand this comment.
Conclusion: Should be improved to highlight main results and respective implications for research and policy.
Answer from the authors: We have rewritten the conclusion as follows: The average price (per 100 kcal) for UPF was significantly cheaper than for MPF. The contribution of MPF to the daily dietary cost was significantly higher for individuals with higher household education level compared to those with lower household education level Diets with a larger share of UPF are significantly cheaper than those with a lower share of these products while the opposite is found for MPF. Policies that reduce improve relative affordability and accessibility of MPF are recommended.
Reviewer 2 Report
The manuscript presents an important problem regarding the relationship between the price and the choice of food. I found this article valuable, and above all, I appreciate the material collected for its preparation. Nevertheless, I have a few comments. It seems that the study deals with both UPF and MPF, which should be included in the title, the aim of the study, but also in the discussion of the results. At the moment, the discussion of the results is insufficient.
Other more detailed comments are as follows:
Abstract
Lines 13-14. The analyses concerned not only the UPF, but also the MPF. So it can be put in the aim of the study.
Lines 23-24. “The contribution of MPF to the daily dietary cost was significantly higher for individuals with higher household education level compared to those with lower household education level (p<0.01)”. The manuscript is about the UPF share, so information on this type of food would be more appropriate. If the aim is changed then this comment will be invalid.
Introduction
Lines 64-65. It is important to consider both the actual price and the perceived price as determinants of food choice. Even if the UPF price is lower than the MPF, it is interesting to see how this is perceived by consumers. This issue may be discussed in Discussion section.
The introduction did not explain why this issue is worth investigating. We get information that UPF consumption has been shown to be high, and that UPF contributes to many diseases. However, I do not know why the cost of the diet was studied in terms of age (this is written in the title of manuscript, and aim of the study). In my opinion, the price of the diet should be examined in terms of household income.
Materials and methods
Line 116 “household educational level” requires an explanation – whose education? - in my country it refers to the person who has the largest share in creating the household budget
Lines 169 – 171. “The average (over two interview/record days) total daily cost (€/d) as well as the total cost per 2000 kcal (€/2000 kcal) was calculated for different age, sex and household education level groups”. Why was the total cost per 2000 kcal calculated? After all, different age groups have different energy needs.
Lines 171-173. Why contributions of UPF and unprocessed/minimally processed food products to the total daily cost were calculated using data only from the first interview (or record) day of the FCS. I think both days should be included (mean value). In addition, the proportion of daily energy intake from UPF and MPF was calculated from the average over two interview/record days using the FCS 2014/15
Lines 175-177 Why authors compare the proportion of daily energy intake from UPF and MPF between households with different education levels. If I understand correctly, the different types of households, due to education, were represented by various people, i.e. children, adolescents and adults. The comparison would be more relevant if the analyzes were carried out separately in the group of children, adolescents and adults, as it was done in the case of gender.
Results
The age group is internally very diverse in terms of the energy value of the diet. It is a pity that the authors do not provide information on the caloric value of the diet (mean value, range). Then the interpretation of the cost of the diet per 2000 calories would be easier.
Table 1. The range of values should be represented by two numbers (min and max) and additionally the difference (max-min) may be indicated.
Figure 1. In my opinion, the way the results are presented on the graph should be changed to facilitate their interpretation. As in the table above, grouping the results should first include age groups, then food types, and finally gender. You can then easily compare men and women of a certain age group to a given type of food, such as UPF.
Figure 2. There is grouping by age, but then I suggest, as before, the type of food and finally education. It will then be possible to easily compare households with different education from a specific age group in relation to a given type of food, such as UPF.
Line 218 – 0.33 Euro instead of 33.
Discussion
Lines 237-242 These are results rather than a discussion. For example, it is worth trying to explain why “for UPF, no such differences among education level groups were found”.
Lines 243-253 This is not a discussion, but rather an introduction.
Lines 253-255 ‘Based on the results from this study, pricing policies would be recommended to reduce the relative affordability of diets with a high caloric share of UPF and improve relative affordability of diets with a high caloric share of MPF in Belgium”. How may it be achieved? Is a reduction in MPF prices possible, or maybe an increase in tax for UPF? What about low-income households? Please comment on this statement.
The limitations should take into account the lack of information about income and its inclusion in the analysis. Is household education level in Belgium related to income - it is worth informing about it in the discussion section.
Conclusions
I suggest preparing conclusions in more detail. After all, the results concerning gender, age groups and household education were obtained
Author Response
Answer comments reviewers
Reviewer 2
The manuscript presents an important problem regarding the relationship between the price and the choice of food. I found this article valuable, and above all, I appreciate the material collected for its preparation. Nevertheless, I have a few comments. It seems that the study deals with both UPF and MPF, which should be included in the title, the aim of the study, but also in the discussion of the results. At the moment, the discussion of the results is insufficient.
Answer from the authors: Thanks for your comments
Other more detailed comments are as follows:
Abstract
Lines 13-14. The analyses concerned not only the UPF, but also the MPF. So it can be put in the aim of the study.
Answer from the authors: We have now included MPF as part of the aims of the study
Lines 23-24. “The contribution of MPF to the daily dietary cost was significantly higher for individuals with higher household education level compared to those with lower household education level (p<0.01)”. The manuscript is about the UPF share, so information on this type of food would be more appropriate. If the aim is changed then this comment will be invalid.
Answer from the authors: As per your previous comment, the aim of the study has been changed.
Introduction
Lines 64-65. It is important to consider both the actual price and the perceived price as determinants of food choice. Even if the UPF price is lower than the MPF, it is interesting to see how this is perceived by consumers. This issue may be discussed in Discussion section.
Answer from the authors: This is important but there is not really any literature on this, I think it depends on the types of consumers on whether the difference in cost is really a barrier or not.
The introduction did not explain why this issue is worth investigating. We get information that UPF consumption has been shown to be high, and that UPF contributes to many diseases. However, I do not know why the cost of the diet was studied in terms of age (this is written in the title of manuscript, and aim of the study). In my opinion, the price of the diet should be examined in terms of household income.
Answer from the authors: We have now updated the aims to more generally study the cost of diets according to their caloric shares of UPF and MPF in Belgium. We agree that it would be interesting to study the price of the diet in terms of household income, but the Belgian food consumption survey did not collect data on household or personal income, only on household and individual education level. We added this as a limitation of the study.
Materials and methods
Line 116 “household educational level” requires an explanation – whose education? - in my country it refers to the person who has the largest share in creating the household budget
Answer from the authors: It is the education level of the person with the highest education level within the household of the participant. We have now clarified this in the manuscript.
Lines 169 – 171. “The average (over two interview/record days) total daily cost (€/d) as well as the total cost per 2000 kcal (€/2000 kcal) was calculated for different age, sex and household education level groups”. Why was the total cost per 2000 kcal calculated? After all, different age groups have different energy needs.
Answer from the authors: We looked at both the total cost per day as well as per 2000 kcal. The price per 2000 kcal is also included as different age or gender groups may consume different types of foods in different quantities which can result in diets for certain groups being more expensive or cheaper for the same amount of energy.
Lines 171-173. Why contributions of UPF and unprocessed/minimally processed food products to the total daily cost were calculated using data only from the first interview (or record) day of the FCS. I think both days should be included (mean value). In addition, the proportion of daily energy intake from UPF and MPF was calculated from the average over two interview/record days using the FCS 2014/15
Answer from the authors: We actually did use the 2 24-hour recalls (or records for children) in all analyses. We have updated the data analysis section to clarify this.
Lines 175-177 Why authors compare the proportion of daily energy intake from UPF and MPF between households with different education levels. If I understand correctly, the different types of households, due to education, were represented by various people, i.e. children, adolescents and adults. The comparison would be more relevant if the analyzes were carried out separately in the group of children, adolescents and adults, as it was done in the case of gender.
Answer from the authors: We have carried out these analysis by age group as they are presented in Figure 2. We have updated the data analysis section to clarify this.
Results
The age group is internally very diverse in terms of the energy value of the diet. It is a pity that the authors do not provide information on the caloric value of the diet (mean value, range). Then the interpretation of the cost of the diet per 2000 calories would be easier.
Answer from the authors: We print here below the results on kcal/day intake for the different age groups by gender. We however do not feel it is really necessary to add those to the manuscript. We look at both total cost per day and cost per 2000 kcal per day.
|
sex |
age |
mean |
95% CI |
N |
|
males |
3-5 years |
1425 |
(1377-1459) |
230 |
|
6-9 years |
1761 |
(1726-1811) |
279 |
|
|
10-13 years |
2008 |
(1962-2062) |
210 |
|
|
14-17 years |
2165 |
(2122-2222) |
240 |
|
|
18-39 years |
2341 |
(2264-2397) |
305 |
|
|
40-64 years |
2223 |
(2151-2298) |
284 |
|
|
females |
3-5 years |
1341 |
(1296-1394) |
224 |
|
6-9 years |
1552 |
(1509-1586) |
259 |
|
|
10-13 years |
1639 |
(1596-1684) |
239 |
|
|
14-17 years |
1678 |
(1643-1729) |
239 |
|
|
18-39 years |
1702 |
(1657-1748) |
315 |
|
|
40-64 years |
1598 |
(1553-1636) |
322 |
Table 1. The range of values should be represented by two numbers (min and max) and additionally the difference (max-min) may be indicated.
Answer from the authors: We have actually deleted the SE and the range from the Table 1 and the Table 2 in Annex and have now given the confidence intervals which are much more informative.
Figure 1. In my opinion, the way the results are presented on the graph should be changed to facilitate their interpretation. As in the table above, grouping the results should first include age groups, then food types, and finally gender. You can then easily compare men and women of a certain age group to a given type of food, such as UPF.
Answer from the authors: We have now changed this figure according to your suggestion
Figure 2. There is grouping by age, but then I suggest, as before, the type of food and finally education. It will then be possible to easily compare households with different education from a specific age group in relation to a given type of food, such as UPF.
Answer from the authors: We have now changed this figure according to your suggestion
Line 218 – 0.33 Euro instead of 33.
Answer from the authors: This has been changed, since we also updated this analysis, the value is slightly different as well.
Discussion
Lines 237-242 These are results rather than a discussion. For example, it is worth trying to explain why “for UPF, no such differences among education level groups were found”.
Answer from the authors: We have added the explanation that is due to the fact that there are no significant differences among education level groups in the consumption of UPF, only in the consumption of MPF, as was highlighted in a previous Belgian study. We have also added references to some other studies which do find a reverse association between consumption of UPF and socioeconomic position.
Lines 243-253 This is not a discussion, but rather an introduction.
Answer from the authors: We prefer to keep those sentences on policy options in other countries as part of the discussion rather than the introduction. We have removed however the following as it does not fit very well with the discussion: NOVA is now recognized as a valid tool for nutrition and public health research, policy and action, in reports from the Food and Agriculture Organization (FAO, 2015) of the United Nations and the Pan American Health Organization (PAHO, 2016).
Lines 253-255 ‘Based on the results from this study, pricing policies would be recommended to reduce the relative affordability of diets with a high caloric share of UPF and improve relative affordability of diets with a high caloric share of MPF in Belgium”. How may it be achieved? Is a reduction in MPF prices possible, or maybe an increase in tax for UPF? What about low-income households? Please comment on this statement.
Answer from the authors: There are plenty of countries which already introduced taxes on certain groups of UPF foods in order to discourage consumption, while a few others have also reduced the price of fresh fruits and vegetables, such as in Australia for example where the Goods and Services Tax does not apply to these foods. It is all a matter of political will but it certainly is possible. We have added a sentence however that improving accessibility to MPF in other ways is also needed: Also, other structural non-fiscal interventions (i.e. whole food reformulation rather than nutrient-specific reformulation, food procurement policies) will be required to improve accessibility to convenient and affordable minimally processed foods and meals (Adams et al, 2020).
The limitations should take into account the lack of information about income and its inclusion in the analysis. Is household education level in Belgium related to income - it is worth informing about it in the discussion section.
Answer from the authors: We have now added this limitation (the lack of data on income) in the discussion section
Conclusions
I suggest preparing conclusions in more detail. After all, the results concerning gender, age groups and household education were obtained
Answer from the authors: We have updated the conclusion section as follows: We have rewritten the conclusion as follows: The average price (per 100 kcal) for UPF was significantly cheaper than for MPF. The contribution of MPF to the daily dietary cost was significantly higher for individuals with higher household education level compared to those with lower household education level Diets with a larger share of UPF are significantly cheaper than those with a lower share of these products while the opposite is found for MPF. Policies that reduce improve relative affordability and accessibility of MPF are recommended.
Round 2
Reviewer 1 Report
Dear authors,
I appreciate your kind responses to the inquiries. I would like to congratulate you for the improvements in the manuscript.
There are two points in my perspective to be clarified.
1) What is the rationale for stratified analysis by sex and age? There is no explanation in the introduction, and the results were not further discussed.
2) Considering that MPF are consumed with culinary ingredients in the form of meals, what is the rationale to not compare the cost of UPF diets with the price per calories of the MPF and processed culinary ingredients accounted together?
Thank you very much.
Author Response
REVIEWER1
Dear authors,
I appreciate your kind responses to the inquiries. I would like to congratulate you for the improvements in the manuscript.
Answer from the authors: Thank you very much
There are two points in my perspective to be clarified.
1) What is the rationale for stratified analysis by sex and age? There is no explanation in the introduction, and the results were not further discussed.
Answer from the authors: As you can see from appendix table 1 (and now explained in the first paragraph of the results section) significant differences between males and females were found for the % energy consumed from MPF while significant differences were found between age groups for the % energy consumed from UPF. We had specified in introduction and objectives (which was changed from the previous version) that we stratified by different socio-demographic groups, including education level, age and sex. We do include all strata (sex/age group/education level) into the discussion of the results.
2) Considering that MPF are consumed with culinary ingredients in the form of meals, what is the rationale to not compare the cost of UPF diets with the price per calories of the MPF and processed culinary ingredients accounted together?
Answer from the authors: We prefer to analyze them separately, according to the original groups as part of the NOVA classification. In reality UPF or processed foods can also be consumed with culinary ingredients, i.e. adding extra salt or sugar or oil for example. In addition, we want to support policies that specifically stimulate the consumption of MPF and reduce the consumption of UPF. For processed foods or processed culinary ingredients concrete policy actions are less clear.
Reviewer 2 Report
Thank you for taking my comments into account.
Kind regards
Reviewer
Author Response
Thank you very much